# Study on the Preparation and Properties of Talcum-Fly Ash Based Ceramic Membrane Supports

**DOI:** 10.3390/membranes10090207

**Published:** 2020-08-28

**Authors:** Chao Cheng, Hongming Fu, Jun Wu, Heng Zhang, Haiping Chen

**Affiliations:** 1School of Energy, Power and Mechanical Engineering, North China Electric Power University, Beijing 102206, China; chengc@ncepu.edu.cn (C.C.); Fuhm@ncepu.edu.cn (H.F.); hdchp@ncepu.edu.cn (H.C.); 2Wusu Thermal Power Branch of Xinjiang Energy Chemical Co., Ltd. of State Power Investment Corporation Limited, Wusu 833300, China; wujun0106@126.com

**Keywords:** fly ash, ceramic membranes support, water recovery, preparation

## Abstract

Ceramic membrane method for moisture recovery from flue gas of thermal power plants is of considerable interest due to its excellent selection performance and corrosion resistance. However, manufacturing costs of commercial ceramic membranes are still relatively expensive, which promotes the development of new methods for preparing low-cost ceramic membranes. In this study, a method for the preparation of porous ceramic membrane supports is proposed. Low-cost fly ash from power plants is the main material of the membrane supports, and talcum is the additive. The fabrication process of the ceramic membrane supports is described in detail. The properties of the supports were fully characterized, including surface morphology, phase composition, pore diameter distribution, and porosity. The mechanical strength of the supports was measured. The obtained ceramic membrane supports displays a pore size of about 5 μm and porosity of 37.8%. Furthermore, the water recovery performance of the supports under different operating conditions was experimentally studied. The experimental results show that the recovered water flux varies with operating conditions. In the study, the maximum recovered water flux reaches 5.22 kg/(m^2^·h). The findings provide a guidance for the ceramic membrane supports application of water recovery from flue gas.

## 1. Introduction

Currently, China is suffering serious water shortages and water pollution problems caused by, among others, rapid economic development [1]. In the industrial sector, thermal power plants consume a vast amount of water, which accounts for about 11% of China’s total industrial water consumption [2]. In short, China’s water resources are limited with a great demand for water and water conservation. In the case of air-cooled units alone in thermal power plants, water consumption of the wet desulfurization system accounts for more than 50% of the total water consumption of the units. However, water vapor discharged into the atmosphere along with the purified flue gas through the chimney accounts for more than 80% of the water consumption of those systems, with a large amount of this water vapor being generated in the coal combustion process. Moreover, a large amount of heat is discharged with flue gas. The content of water vapor in the flue gas discharged by a coal-fired power plant is 4–13%, while that discharged by a gas-fired power plant is more than 20%. A substantial amount of water can be conserved by recovering water vapor from flue gas [3]. Therefore, thermal power plants have huge water conservation potential [4].

Recovering water vapor and its gasification latent heat has attracted great interest in the scientific community over the last few years due to high moisture content in the flue gas of thermal power plants. General methods to recover moisture and its gasification latent heat include the condensation method and solution absorption method. The condensation method mainly adopts an indirect condensation heat exchanger [5], such as a low temperature economizer [6], a low pressure economizer [7,8], and a direct contact condenser [9]. However, acid gas SO_2_ exists in the flue gas that condense on the surface of the heat exchanger at lower temperature than the acid dew point, which causes low-temperature corrosion. This seriously affects the service life and water recovery performance of the heat exchanger. The solution absorption method employs a solution with moisture absorption behavior as a desiccant, such as lithium bromide, lithium chloride, and calcium chloride solution, which directly contacts with flue gas and promotes the removal of flue gas moisture. Westerlund et al. [10] adopted a two-stage absorption cycle system to remove flue gas particles and recycle the waste heat in a biomass boiler. The absorber adopted the packing type, which was divided into three stages to heat the return water using the recovered heat. At the same time, some hot flue gas is used for solution regeneration.

Considering the limitations of condensation heat exchanger that is susceptible to corrosion, the moisture recovery method of flue gas based on the organic fiber membrane is preferred. The organic composite membrane is superior due to low production cost, large surface area per unit volume, low weight, small floor area, and easy modularization. Chen et al. [11] prepared hydrophilic composite hollow fiber membrane sulfonated polyether ether ketone-polyether sulfone (SPEEK-PES) that possessed excellent thermal stability and mechanical properties. They also experimentally studied the effect of scavenging the temperature and flow rate on membrane performance using simulated flue gas. The result showed that water recovery efficiency increased with the rise in temperature. Choi et al. [12] prepared a thin-film composite (TFC) hollow fiber membrane via polymerization of 3,5-diaminobenzoic acid (DABA) aqueous solution and trimethyl chloride on the inner surface of polysulfone (PSF) hollow fiber. The obtained membranes were used to create membrane modules, and the performance of the membrane module for recovering water under different operating conditions was studied experimentally. The maximum water flux of the membranes was 2.3 kg/(m^2^·h), and the water recovery efficiency was 27.7%. Volkov et al. [13] used the polymer membrane to construct a porous condenser. In the experiment, the permeate was used as the coolant to conduct membrane distillation for desalination of NaCl solutions and thermopervaporation for removal of n-butanol from the aqueous solution. The results showed that the water flux in the membrane distillation process of the porous condenser could reach 21 kg/(m^2^·h). In addition, Borisov et al. [14] combined the porous condenser thermopervaporation and liquid–liquid phase separation method, and used it for the first time to recover and concentrate butanol in the fermentation broth, and analyzed the permeation flux of butanol and water. Although the polymer membrane has a certain separation performance, it can also be used to recover water from flue gas. It is difficult to apply organic composite membranes to practical industry processes on large scale due to their low mechanical strength.

Compared with the polymer membrane, inorganic ceramic composite membranes possess good mechanical properties to adhere to industrial requirements [15], and enhanced thermal stability, chemical stability, and corrosion resistance of ceramic membranes by ensuring their capability with high temperature and a wide pH range [16]. In addition, the pore size of the ceramic membrane selection layer can be adjusted to compliment certain requirements by promoting excellent selectivity [17]. Wang et al. [18] proposed the recovering of waste heat and water from flue gas using nano-porous ceramic membranes, according to the capillary condensation principle. An engineering demonstration was carried out for industrial boiler flue gas waste heat and moisture recovery. The steam recovery efficiency reached 40% by achieving good recovery performance. Chen et al. [3,19] studied the recovery performance of nanofiltration ceramic membranes of varied pore sizes under different experimental conditions. The ceramic membrane with a 20-nm pore size is suitable for different flue gas conditions with the highest water recovery efficiency of 55%. Gao et al. [20] designed a membrane module consisting of 46 porous ceramic membranes with a pore size of 1 μm and length of 40 cm. They studied its water and heat recovery performance in the flue gas from gas-fired boiler under different conditions, and maximum recovered water flux of 15.77 kg/(m^2^·h).

Although ceramic membranes have excellent performance in flue gas moisture recovery, high production costs remain a major obstacle for large-scale application. Conventional ceramic membranes consisting of pure Al_2_O_3_, TiO_2_, ZrO_2_, SiO_2_, or mixtures [15,21,22] require sinter temperature above 1500 °C. Expensive raw materials and large energy consumption during sintering results in high manufacturing cost of ceramic membranes [23,24]. At present, reports have shown the preparation of low-cost porous ceramic membrane using cheap traditional ceramic materials such as clay and kaolin. Elgamouz et al. [25] employed natural clay minerals as raw materials to prepare flat porous ceramic membrane supports of varied pore size by sintering at different temperatures, and coated them with tetra-ethyl orthosilicate by hydrothermal deposition. Mohammed et al. [26] used clay and AlF_3_·3H_2_O as raw materials to synthesize mullite-based porous ceramic membranes at 1400 °C sintering temperature. Hou et al. [27] added MoO_3_ to the raw materials consisting of kaolin and Al_2_O_3_ to prepare a mullite-based ceramic membrane with a porosity of 67% and shrinkage of −2.57% at 1400 °C sinter temperature.

Fly ash is a by-product of thermal production in power plants, which is currently employed in the cement industry [28]. However, the application of fly ash for ceramic membrane production is still in its infancy. Fang et al. [29] prepared tubular ceramic membrane supports with fly ash of a different particle size distribution with a coated selective layer. The average pore size and pure water flux obtained for the ceramic membrane were 0.77 μm and 1.56 × 10^4^ L/(m^2^·h·bar), respectively. Zhu et al. [30] produced porous ceramic membranes with the main crystalline phase of mullite using fly ash as the raw material with the addition of a different mass fraction of Al_2_O_3_. Qin et al. [31] applied ceramic membranes prepared with fly ash as precursor in juice clarification. Suresh et al. [32] employed fly ash, quartz, and calcium carbonate to fabricate ceramic microfiltration membranes and evaluated membrane performances using synthetic oil-water emulsions. Zou et al. [33] used fly ash to prepare a ceramic microfiltration membrane and used in oil-water emulsion treatment. Fu et al. [34] fabricated mullite ceramic membranes by using fly ash and Al(OH)_3_ and MoO_3_ and planned for water filtration. The above research mainly focused on studying the influence of different factors on membrane preparation and the application in juice clarification, oil-water treatment, and water filtration, but less in flue gas water recovery.

In this study, low-cost fly ash-based porous ceramic membrane supports were prepared and studied. The fly ash is the main raw material, and talc powder is the additive. First, the supports were characterized by using scanning electron microscopy to observe the surface morphology of membrane supports, and utilizing mercury porosimeter to measure the pore diameter distribution. Besides, the mechanical strength was measured. Then, the water recovery performance of the fly ash-based porous ceramic membrane supports from flue gas under different conditions (e.g., flue gas flow rate, flue gas temperature, cooling water flow rate, and cooling water temperature) was experimentally studied. This article dedicates to prepare and study the low-cost ceramic membrane supports, and provides a guidance for flue gas moisture recovery application of ceramic membrane supports.

## 2. Experimental

### 2.1. Membrane Preparation

Ceramic membrane supports were prepared using fly ash, which was produced by power plants, as the main material and talcum as the sintering aid. Table 1 lists the main chemical composition of ceramic membrane supports of raw materials. Carboxymethylcellulose and dextrin as the binder, and glycerin as the plasticizer were mixed with water in a certain proportion, and then added to the mixture of fly ash and talcum to form mud embryos, which were sintered into ceramic membrane supports. The specifications of raw materials used in this experiment are summarized in Table 2.

In this paper, ceramic membrane supports were prepared by the extrusion method, which is a common method of tubular ceramic membrane preparation [35,36,37,38]. The specific preparation process of the support is as follows. The pretreated fly ash, talc powder, and other ceramic additives were mixed by a mixer (HL-5, Hangzhou, China) and stirred uniformly with an appropriate amount of water. The pretreatment of fly ash included grinding and screening. First, ball mill (XQM-20, Changsha, China) was used to grind for 30 min, and the large particles in fly ash raw materials were crushed. Then the 80-mesh sieve was used on the sieving machines to screen for 10 min. The purpose of pretreatment was mainly to remove large particles of fly ash and impurities, so that the size of fly ash particles was uniform and the particle size distribution was narrow, which ensured the preparation of ceramic membrane supports with a uniform pore size and stability. As a sintering aid, talc powder could be used as a magnesium source supplement due to its high magnesium content, which could make the density uniform and surface smooth of the membrane supports after sintering.

In the process of vacuum pugging, the mixed mud was put into the vacuum chamber of the vacuum pugging machine (S-48, Hebi, China), and kept the relative vacuum pressure at −35 kPa. Under the action of the pressure difference between the air bubble and the vacuum chamber, the air bubble broke and was discharged out of the vacuum chamber. After being refined 2–3 times, the mud was placed at 30 °C in an opaque and ventilated environment for 24 h. Then, the green bodies were refined for 3–4 times again to obtain sufficient strength and shape. Vacuum pugging was used to eliminate the residual air in the ceramic slurry, and ensure that the components were evenly mixed by avoiding the appearance of large pore defects in the ceramic membrane supports during the sintering process. Through stalling, the moisture in the slurry was more uniform while the strength of green bodies and the molding performance of the slurry were improved.

The wet green bodies of the fly ash-based ceramic membrane were obtained by extruding the green bodies using an extruder (LWJ63, Hebi, China). The wet green bodies without deformation or crake were aligned by a straightening machine (XZJ-25/1000, Hebi, China) and dried for 48 h, generating the green body of the fly ash-based tubular ceramic membrane supports during the process. Since the supports extruded by the extruder was in the form of a wet green body, the extrusion process could make the supports body bend. Therefore, it is necessary to use a straightening machine for straightening and drying to ensure the straightness of the supports.

Then, under the premise of ensuring that the sintering does not crack and the layer is continuous, the tubular fly ash-based ceramic membranes were obtained by sintering at appropriate sintering parameters in a high temperature pipe furnace (TSK-8-14). After drying, sintering was carried out according to the predetermined sintering temperature curve. At last, the membrane supports are pretreated and cleaned with deionized water to remove water-soluble impurities. The concrete operational process is shown in Figure 1.

### 2.2. Characterization Method

The particle size of fly ash used in this experiment was measured by Laser Scattering Particle Size Distribution Analyzer (LA-960, Horiba, Nagoya, Japan). X-ray diffraction (XRD, AL-Y3000, Aolong, Dandong, China) was used to analyze the phase composition of fly ash and the phase composition of the ceramic membrane supports. To determine the appropriate sintering parameters, the green bodies were analyzed by a thermogravimetric analyzer (TGA-1450A, Innuo, Shanghai, China). The surface morphology of the ceramic membrane supports was observed by a scanning electron microscopy (SEM) (JSM6490LV, JEOL, Tokyo, Japan). The pore size and porosity of the support were examined using a mercury porosimeter (Poremaster 60, Anton-Paar, Graz, Austria), and its mechanical strength was measured using the three-point bending method.

The method for measuring the bending strength is shown in Figure 2. Since the sample of 120 mm is tubular and the bending head is a cylinder surface, contact of the intermediate force point is the point contact, which results in a much lower measured bending strength than the actual value.

The bending strength calculation formula is as follows.
(1)$$\sigma =\frac{M\cdot y}{I_{z}}=\frac{\left(1/4\cdot FL\right)\cdot D/2}{\pi \left(D^{4}-d^{4}\right)/64}$$
where σ is the bending strength, MPa, *M* is the moment, N·mm, *I_Z_* is the inertia moment of the section to the z axis, mm^4^, *F* is the load when the sample breaks, N, *L* is the spacing of the support blade, mm, and *D* and *d* are the outer and inner diameter of the sample, mm, respectively.

The corrosion measurement method is as follows. Sulfuric acid solution with a mole fraction of 3 mol/L was prepared using analytically pure sulfuric acid and deionized water. The sample was immersed in the sulfuric acid solution at a slight boiling state for 24 h. After, the sample was washed by deionized water and completely dried at a constant temperature prior to the measurement.

### 2.3. Water Recovery Experiment of Flue Gas

#### 2.3.1. Experimental Devices

The simulated flue gas dehydration experiment was carried out with the self-made fly ash-based ceramic membrane supports. The length, outer diameter, inner diameter, and effective membrane area of the self-made fly ash-based ceramic membrane supports are 80 cm, 12 mm, 8 mm, and 0.029 m, respectively. The experimental system diagram is shown in Figure 3.

As shown in Figure 3, the whole experimental system consists of a flue gas system and cooling water system. The flue gas, composed of nitrogen and water vapor, is prepared artificially. The simulated flue gas is introduced into the shell side of the membrane module and passes through the outside of the membrane tube, while water flows inside the membrane tube for cooling down the tube. Flue gas and water flows counter-currently. In order to avoid heat loss, membrane module housing and pipes are treated with insulating cotton.

The flue gas flow rate is controlled by a mass flow controller. Nitrogen first enters a humidifier of a specific temperature through the mass flow controller, then enters the membrane module, scours the membrane tube, and then discharges to the atmosphere. The humidifier is heated by a water bath with adjustable temperature. The temperature and relative humidity of flue gas at the inlet and outlet of the membrane module are measured by a temperature and humidity transmitter. The relative humidity (RH) of simulated flue gas is about 100%.

In terms of cooling water, water is pumped out of the constant temperature water tank by the self-priming pump flowing through the membrane tube and entering the return tank. The inlet and outlet temperature of cooling water are measured, respectively, by thermocouples set at the inlet and outlet of the membrane tube. A flow meter controlled the cooling water flow rate. The relative vacuum is about −20 kPa in experiments.

Under the action of cooling water, after the hot flue gas enters the membrane module, when flue gas scours the membrane tube with a lower temperature, water vapor condenses on the surface of the membrane tube, and the condensed droplets enter the membrane tube along pores and discharges with the cooling water. The experimental apparatus used in the experiment are tabulated in Table 3.

#### 2.3.2. Experimental Performance Evaluation

The application of the membrane in the field of wet gas flow includes dehydration and water recovery. For dehydration, the dry gas flux was analyzed in References [39,40]. In terms of water recovery from flue gas, the recovered water flux is usually taken as the index, which could be found in References [18,41,42,43,44]. These research studies focus on recovering water to solve the problem of water shortage, such as using recovered water for desulfurization to achieve zero water consumption in desulfurization.

This paper focuses on the preparation of low-cost ceramic membrane, which is then used to recover moisture in flue gas. Therefore, the recovered water flux and water recovery efficiency are used as analysis indicators.

The recovered water flux was calculated as follows.
(2)$$J_{rec}=\frac{m_{in}-m_{out}}{S}$$
where $J_{rec}$ is the recovered water flux, kg/(m^2^·h), $m_{in}$ and $m_{out}$ are the water vapor content in flue gas flow at the inlet and outlet of membrane module, respectively, kg/h, and *S* is the membrane area, m^2^.

The water recovery efficiency was given by the equation below.
(3)$$\eta_{w}=1-\frac{m_{out}}{m_{in}}$$
where $\eta_{w}$ is the water recovery efficiency (%).

In a laboratory-scale study, the energy consumption by the system is only the power consumption of the water pump that provides cooling water. The energy consumption is related to the membrane area. According to the experimental results, the power consumption of the water pump is 0.53 (kw·h)/m^2^.

### 2.4. Uncertainty Analysis

The testing uncertainty may cause experimental errors. Uncertainty analysis was performed in order to preserve the accuracy of experimental results in the study. The direct testing parameters contain a flue gas flow rate *Q_N2,in_* and *Q_N2,out_*, relative humidity *φ_in_* and *φ_out_*, the load *F*, and the spacing of the support blade *L*.

The relative uncertainty of the bending strength Δ*σ* was given by the equation below.
(4)$$\Delta \sigma =\frac{\sqrt{{\left(\frac{\partial \sigma }{\partial F}\Delta F\right)}^{2}+{\left(\frac{\partial \sigma }{\partial L}\Delta L\right)}^{2}}}{\sigma }$$

The relative uncertainty of the recovered water flux Δ*J_w_* was computed from Equation (5).
(5)$$\Delta J_{w}=\frac{\sqrt{{\left(\frac{\partial J_{w}}{\partial Q_{N_{2},in}}\Delta Q_{N_{2},in}\right)}^{2}+{\left(\frac{\partial J_{w}}{\partial \varphi_{in}}\Delta \varphi \right)}^{2}+{\left(\frac{\partial J_{w}}{\partial \varphi_{out}}\Delta \varphi \right)}^{2}+{\left(\frac{\partial J_{w}}{\partial Q_{N_{2},out}}\Delta Q_{N_{2},out}\right)}^{2}}}{J_{w}}$$

Through calculation, the maximum relative uncertainty of the bending strength before and after corrosion is 3.42%, and the maximum relative uncertainty of the recovered water flux Δ*J_w_* is 3.56%.

## 3. Results and Discussion

### 3.1. Characterization of Raw Materials

A SEM image of fly ash after pretreatment is shown in Figure 4. The particle size distribution of fly ash is between 1–10 μm with an average particle size of 3.29 μm. According to the SEM image, most fly ash particles are spherical. Their chemical composition are mainly alumina and silica, which is suitable for the preparation of ceramic membrane supports.

The thermogravimetric analysis of the green body using 10 °C/min heating rate in an air atmosphere is shown in Figure 5. The results indicate that free water is removed at about 100 °C while some bound water is removed at about 200 °C. Decomposition of the binder promotes an exothermic reaction at about 350 °C, which results in a small temperature change in the differential thermal analysis. However, insignificant weight loss is observed at this temperature. Above 700 °C, the differential thermal analysis temperature is greater than that at ambient temperature due to combustion of the remaining carbon particles inside the material and the decomposition of the organic additive. According to the thermal analysis results, the sintering parameters are determined preliminarily. The results are shown in Figure 6.

The thermogravimetric analysis results reveal a clear weight loss of the ceramics at 135 and 350 °C. Thus, it is necessary to adopt heat preservation to ensure complete removal of adsorbed water and combined water. A lower temperature rise rate was adopted between 600 and 900 °C to allow the burnout of pore-forming agents in ceramics. When the temperature reaches 1200 °C, it is maintained for 3 h to increase the densification degree of ceramic crystal and the strength of support.

### 3.2. Characterization of Membrane Supports

The self-made ceramic membrane supports were characterized. Scanning electron microscopy was used to observe the surface morphology of the membrane supports. The XRD diffraction pattern was used to characterize the phase composition of the membrane supports. Mercury porosimeter was used to measure the pore diameter distribution of the membrane supports.

Figure 7 depict scanning electron microscopy images of the surface of fly ash-based ceramic membrane supports under 2000- and 5000-times magnification, respectively. As shown, the grains of the support are uniform and distributed in blocks. The support surface is rough and dense without clear defects.

Figure 8 describes the XRD diffraction pattern of fly ash-based ceramic membrane supports. The results indicate that the main crystal phases of the ceramic membrane supports are anorthite (CaAl_2_Si_2_O_8_, PDF#41-1486) and ringwoodite ((Mg/Fe)_2_SiO_4_, PDF#21-1258). A slight bulge of the diffraction peak at the small angle shows that the material may contain a small amount of amorphous phase components. Cordierite phase is not found in the support, which may be due to the insufficient sintering temperature.

Figure 9 illustrates the mercury adsorption curve of the support, where the pore size of the talcum-fly ash-based ceramic membrane supports is concentrated at 5–10 μm. Figure 10 shows the pore size distribution curve and pore size distribution histogram diagram of talcum-fly ash-based ceramic membrane supports, respectively. Both figures indicate that the pore size of talcum-fly ash-based ceramic membrane supports is relatively concentrated near 5 and 9 μm with a porosity of 37.8%.

It is generally believed that capillary condensation occurs in the membrane tube with a diameter of 2–50 nm. Water vapor is more likely to condense in the pores and prevent other gases from passing through the membrane pores [18,45]. The pore diameter of the support in this paper reaches 5–10 μm. Thus, the water vapor does not undergo capillary condensation in the pores. The gas molecular diameter in the flue gas is in the order of a nanometer. Other condensable gas in the flue gas may condense first and then penetrate into the membrane through the membrane pores. A small amount of non-condensable gas may also enter the membrane under the effect of pressure difference. In this article, the preparation of membrane supports and water recovery in flue gas were studied without coating selective coatings. In the application of flue gas moisture recovery, in order to improve the water quality of the recovered water, a selective layer needs to be coated, which will be implemented in the future membrane preparation research.

The bending strength of the talcum-fly ash-based ceramic membrane supports before and after corrosion is shown in Figure 11, which showed that the excellent bending property sharply decreases to less than 10 MPa after the corrosion test, while the mechanical strength of the supports is about 25 MPa before corrosion. The particulate matter carried by the flue gas causes the membrane tube to wear. The higher the flue gas velocity is, the more serious the membrane tube wear is. The lower mechanical strength of the membrane tube is not conducive to the installation and long-term application of the membrane module. However, the specific values are not reported, and we did not carry out the research in this article. In this paper, the fly ash-based membrane support was prepared. In order to better characterize the performance of the ceramic membrane, the mechanical strength was used as the characterization parameter in the research process. At present, 7 MPa is enough to meet the laboratory experiment requirements.

It can be seen from Table 1 that the main component of fly ash is SiO_2_, Al_2_O_3_, Fe_2_O_3_, and CaO. To the best of our knowledge, however, these components react with sulfuric acid, and the reaction formula is shown as Equations (6) and (7) [46]. Therefore, when sulfuric acid solution is used for corrosion, the prepared supports react with sulfuric acid to generate sulfuric acid mineral salt. In addition, sulfuric acid solution could also cause the degradation of the binder and plasticizer. It may be that the sintering temperature is not enough. The degree of densification is not high, and there are many oxide crystals that are not effectively combined to form a stable crystal structure. Consequently, the mechanical strength of the support decreases after sulfuric acid corrosion, which leads to the fracture of the supports during the installation of the membrane module for flue gas moisture recovery, and shorten its service life.
(6)$$xCaO\cdot ySiO_{2}\cdot zH_{2}O+SO_{4}^{2-}+2H_{2}O\to CaSO_{4}\cdot 2H_{2}O+Si{\left(OH\right)}_{4}$$
(7)$$\begin{array}{l} 3CaO\cdot Al_{2}O_{3}\cdot CaSO_{4}\cdot 12H_{2}O+2CaSO_{4}\cdot 2H_{2}O+16H_{2}O\\ \to 3CaO\cdot Al_{2}O_{3}\cdot 3CaSO_{4}\cdot 32H_{2}O \end{array}$$

### 3.3. Water Recovery Performance Analysis

Figure 12 shows the effect of the flue gas flow rate on flue gas water recovery performance. As the flue gas flow rate increases, the recovered water flux also increases, but the water recovery efficiency decreases significantly with an increasing flue gas flow rate. This is because, when the relative humidity of the flue gas remains constant, the water vapor content increases proportionally with the increase of flue gas flow. Thus, the recovered water flux increases with the rise of the flue gas flow rate. However, an increase of flue gas flow leads to elevation of flow velocity, which promotes the water vapor in flue gas to discharge with flue gas before being recovered. Thus, this process decreases water recovery efficiency.

The influence of flue gas temperature on water recovery performance is shown in Figure 13. When flue gas temperature increases from 40 to 70 °C, the recovered water flux and water recovery efficiency increases from 0.74 to 5.22 kg/(m^2^·h) and 64.8% to 81.4%, respectively. As shown in Figure 13, with flue gas temperature increases, the recovered water flux increases in an approximate parabolic form. That is, the higher the flue gas temperature is, the faster the growth rate of recovered water flux is due to the water vapor content increasing exponentially with an increasing flue gas temperature at a certain flow rate [47]. However, the flue gas enthalpy value increases with the increase of flue gas temperature. With the increase of flue gas temperature, the growth rate of water recovery is less than that of water vapor content. When the cooling water flow rate and temperature are fixed, the uncondensed water vapor increases correspondingly. Although the recovered water flux increases more rapidly, the growth trend of water recovery efficiency tends to grow more slowly.

The effect of the cooling water flow rate on water recovery performance is shown in Figure 14. When the flue gas temperature is at 50 °C and the cooling water temperature is 20 °C, an increase in the cooling water flow rate slightly influences the water recovery of the flue gas. When the cooling water flow increases from 0.5 to 2 L/min, the water recovery flux is approximately 1.50 kg/(m^2^·h), while the water recovery efficiency increases slightly from 73.6% to 74.3%. In our previous study, the recovered water flux and water recovery efficiency increase with a growing cooling water flow rate. Nevertheless, when the flue gas flow rate flowing into the membrane module is constant, the continuous increase of cooling water shows an insignificant effect on recovered water flux and water recovery efficiency. The error analysis results also show that the effect of cooling water flow rate on the water recovery performance seems negligible under the experimental conditions in this paper.

Figure 15 displays the effect of cooling water temperature on water recovery performance. When the cooling water temperature increases from 15 to 25 °C, recovered water flux decreased from 1.66 to 1.50 kg/(m^2^·h), and water recovery efficiency decreases from 79.7% to 73.0%. When the cooling water temperature increases, the temperature difference between the cooling water and flue gas decreases, the heat transfer capacity decreases, and the cooling effect deteriorates, which leads to the decrease of the condensation rate of water vapor. Especially when the cooling water temperature exceeds 25 °C, the recovered water flux declines more significantly. A large amount of water vapor is discharged from the membrane module without condensation. As a result, recovered water flux decreases, and the water recovery efficiency accordingly decreases.

### 3.4. Performance Comparison and Cost Analysis

Compared with other studies, the ceramic membranes manufactured by fly ash [29,31], kaolin [22], and clay [26] are mostly used for juice clarification [31], oil-water emulsion treatment [33], and are currently less used for flue gas moisture recovery. A comparison of different research studies on water flux and experimental conditions is listed in Table 4. As can be seen, under certain experimental conditions, the recovered water flux of the membrane support can reach 5.22 kg/(m^2^·h), which indicates that the ceramic membrane support has a certain water recovery capacity. Although the diameter of the membrane tube and experimental conditions were different, the recovered water flux of a single tube transport membrane condenser (TMC) had little difference, which is clearly lower than that of tube bundle TMC. The recovered water flux was 15.77 kg/(m^2^·h) in Reference [20]. This is because the ratio of the flow gas flow rate to membrane area (Q/A) reached 2286, which was far higher than that in other studies. Therefore, it is believed that the water recovery could grow when the Q/A is increased for the tube bundle by using membrane supports.

In addition, the content of α-alumina in commercial alumina ceramic membrane manufacturing components is about 90% [48], and the price of α-alumina is 5800 ¥/ton. The content of fly ash in fly ash ceramic membrane manufacturing components manufactured in this article is 70%. The price of fly ash is 60 ¥/ton, which shows that the raw material cost of fly ash ceramic membrane is lower. In addition, the sintering temperature of the fly ash ceramic membrane is usually 1200 °C, and the sintering temperature of the alumina ceramic membrane usually requires up to 1500 °C [49]. Therefore, the sintering energy consumption is naturally high. In summary, the cost of the fly ash ceramic membrane is lower than that of the alumina ceramic membrane.

## 4. Conclusions

In this study, ceramic membrane supports were successfully prepared using fly ash, talc powder, and an appropriate amount of carboxymethyl cellulose and dextrin as raw material, sintering aid, and pore former, respectively. The pore size distribution of the obtained ceramic membranes is near 5 and 9 μm with 37.8% porosity. Anorthite and ringwoodite are the main crystalline phases. The bending strength of the obtained ceramic membranes with excellent bending behavior decreases below 10 MPa after sulfuric acid corrosion. The ceramic membrane supports can be regarded as a macro-porous membrane due to its pore size.

The experimental results of moisture recovery from flue gas using self-made supports show that, with the increase of the flue gas flow rate, recovered water flux increases linearly, while the water recovery efficiency is the opposite. Both recovered water flux and water recovery efficiency increases with a growing flue gas temperature, and decreases with an increasing cooling water temperature. The recovered water flux can reach up to 5.22 kg/(m^2^·h) when flue gas temperature is 70 °C. With the increase of cooling water flow rate, the recovered water flux is almost unchanged.

In the future, improving the corrosion resistance and mechanical strength of the fly ash-based ceramic membrane supports will be the focus of this research. In addition, applying the supports to the field of flue gas moisture recovery in coal-fired power plants is planned to be further explored.

## Figures and Tables

**Figure 1 membranes-10-00207-f001:**
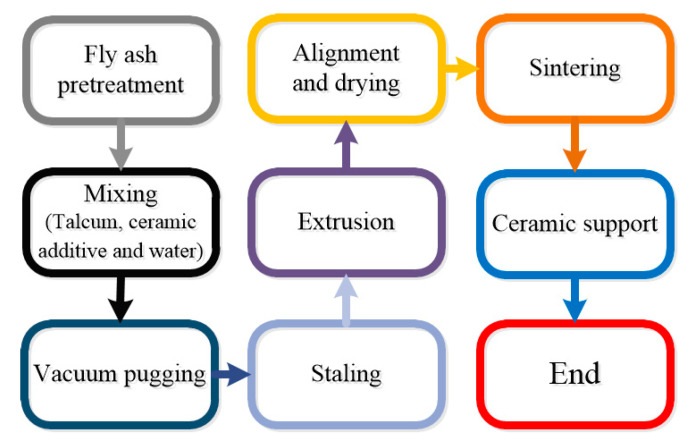
Fly ash-based ceramic membrane supports preparation process.

**Figure 2 membranes-10-00207-f002:**
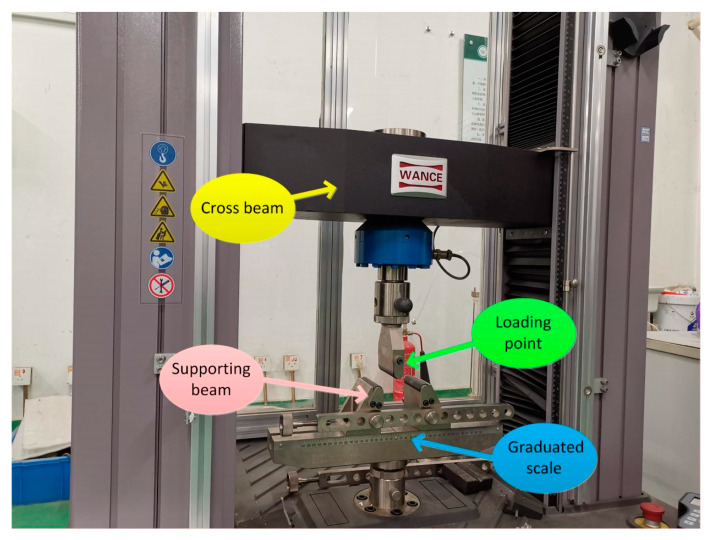
Three-point bending experimental setup.

**Figure 3 membranes-10-00207-f003:**
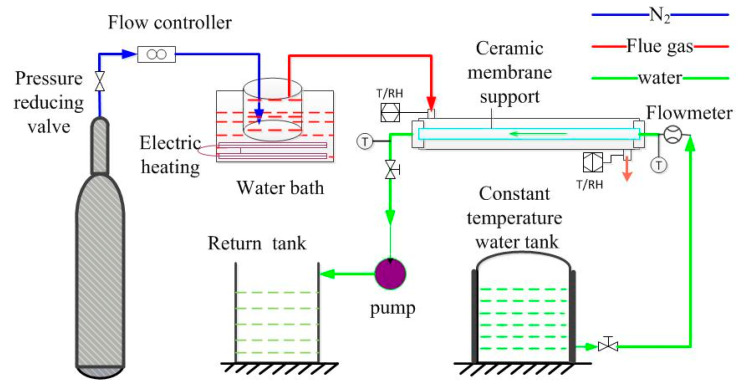
Diagram of the flue gas water recovery experimental system.

**Figure 4 membranes-10-00207-f004:**
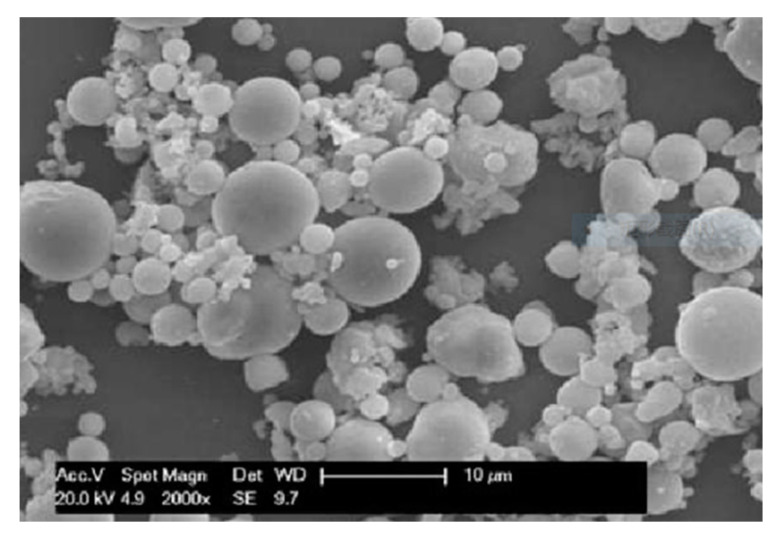
SEM image of fly ash.

**Figure 5 membranes-10-00207-f005:**
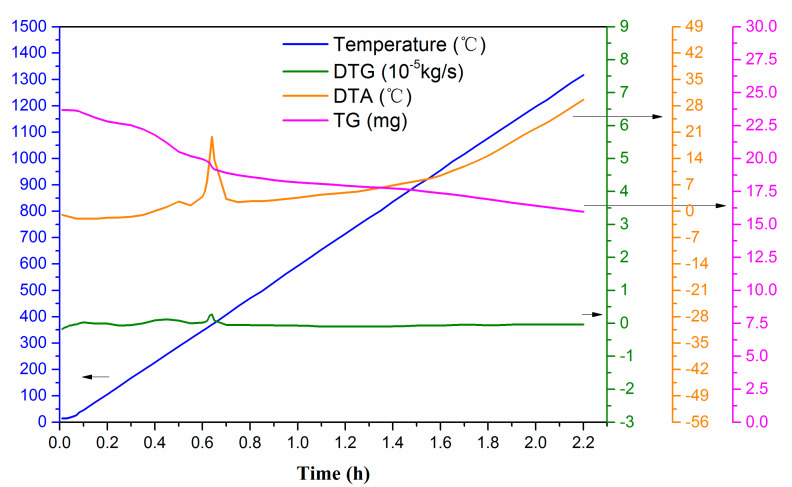
Thermogravimetric analysis curve of the green body.

**Figure 6 membranes-10-00207-f006:**
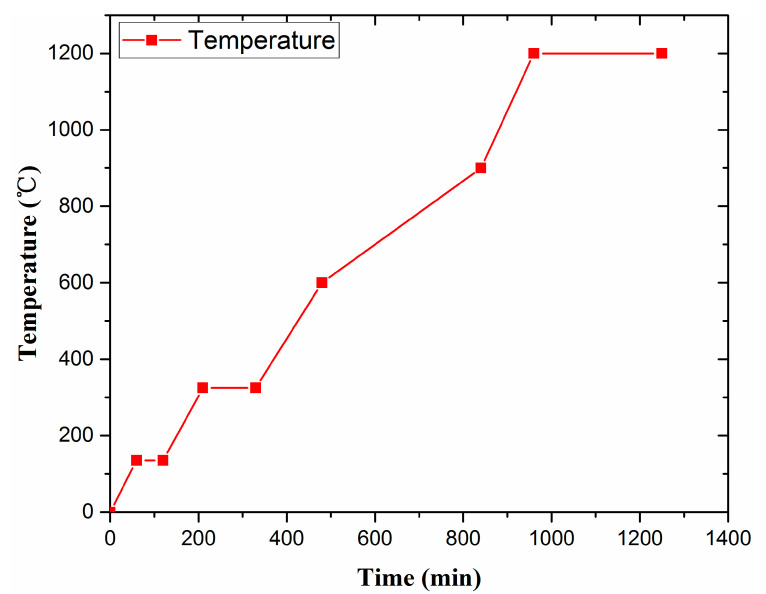
Sintering curve of supports.

**Figure 7 membranes-10-00207-f007:**
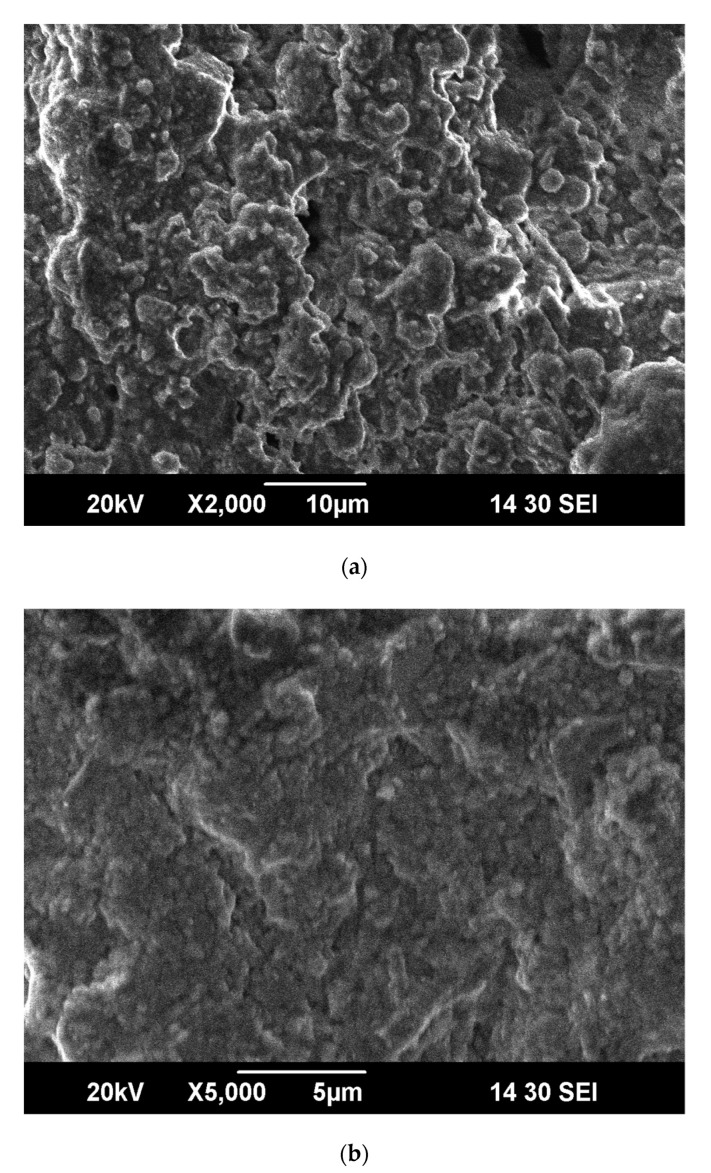
SEM image of support surface magnified 2000 and 5000 times. (**a**) 2000 times. (**b**) 5000 times.

**Figure 8 membranes-10-00207-f008:**
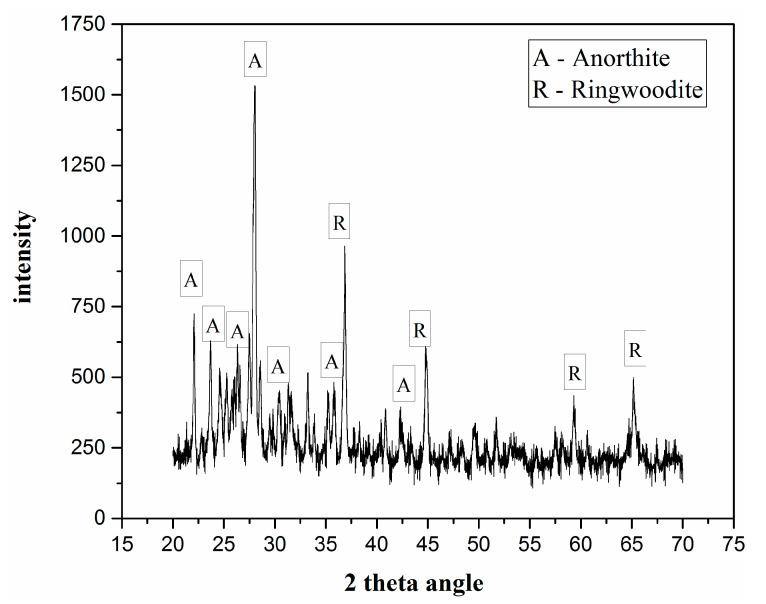
XRD pattern of fly ash-based support.

**Figure 9 membranes-10-00207-f009:**
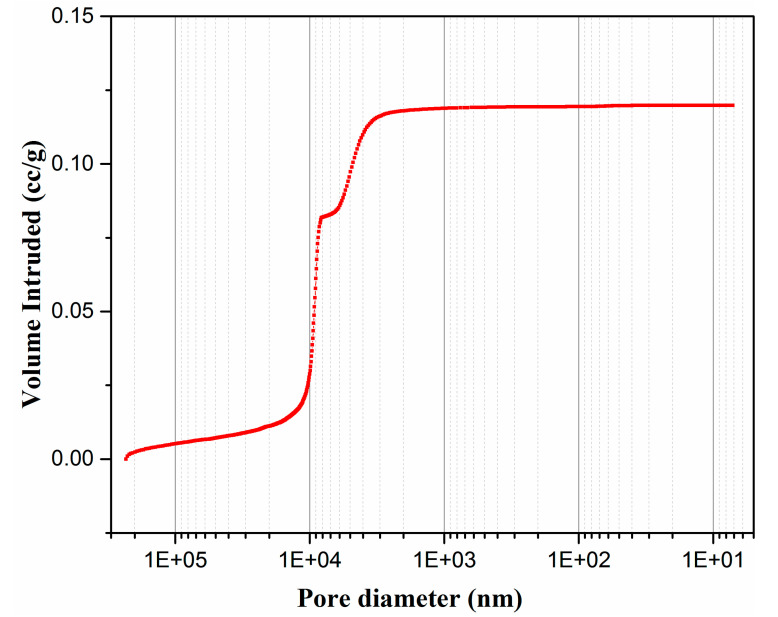
Mercury adsorption curve of talcum-fly ash support.

**Figure 10 membranes-10-00207-f010:**
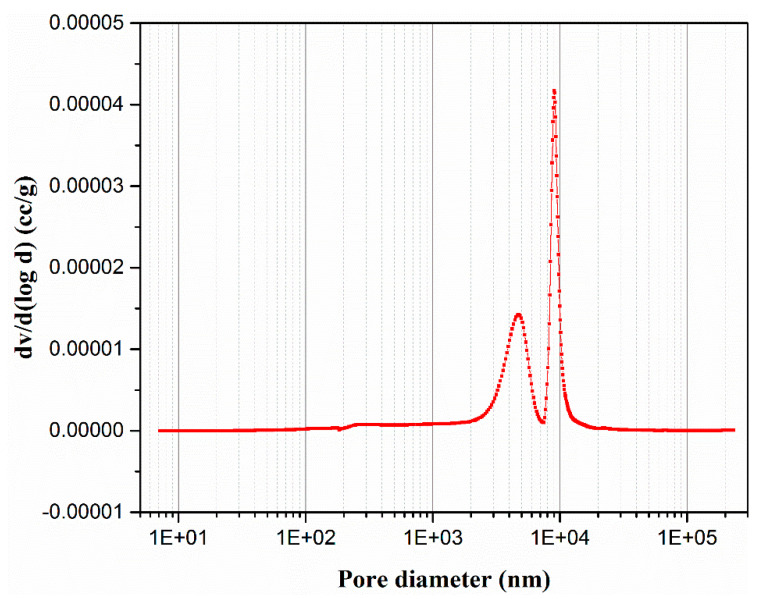
Pore diameter distribution curve of talcum-fly ash support.

**Figure 11 membranes-10-00207-f011:**
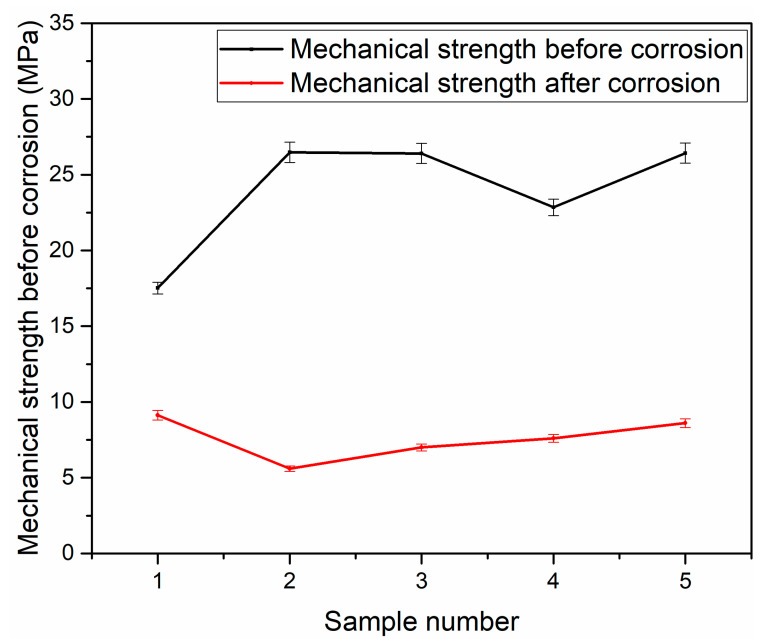
Mechanical strength of support before and after corrosion.

**Figure 12 membranes-10-00207-f012:**
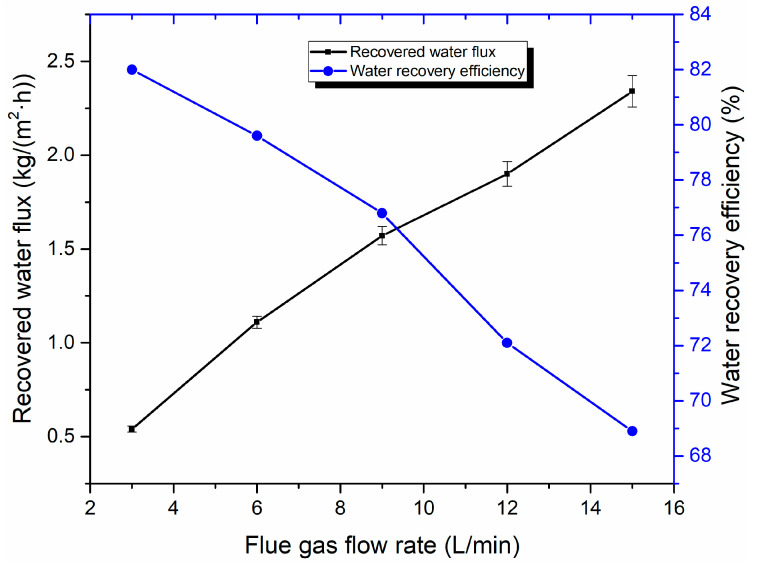
Effect of flue gas flow rate on water recovery performance (Experiment conditions: flue gas temperature 50 °C, cooling water flow rate of 1 L/min, cooling water temperature of 20 °C).

**Figure 13 membranes-10-00207-f013:**
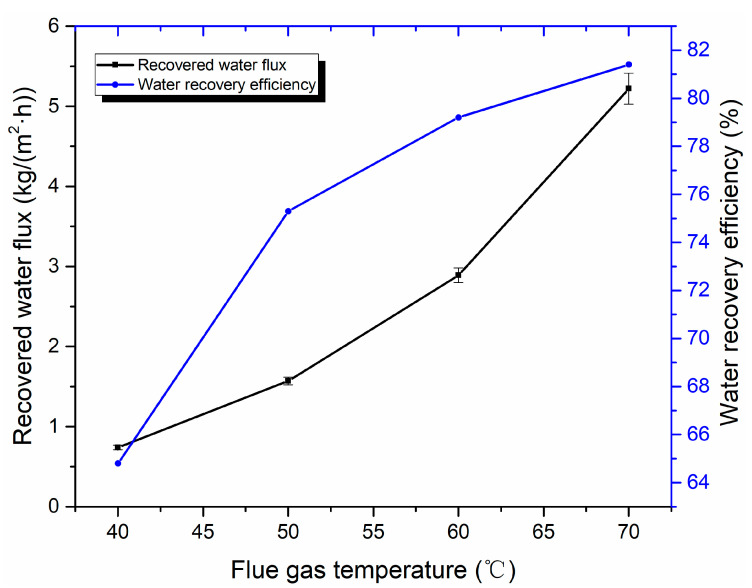
Effect of flue gas temperature on water recovery performance (Experiment conditions: flue gas flow 9 L/min, cooling water flow rate 1 L/min, and cooling water temperature of 20 °C).

**Figure 14 membranes-10-00207-f014:**
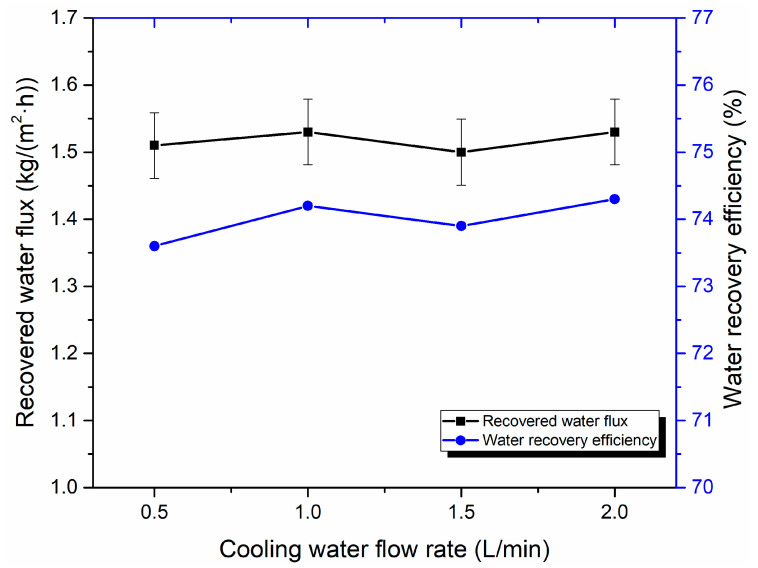
Effect of cooling water flow on water recovery performance (Experiment conditions: flue gas flow rate 9 L/min, flue gas temperature 50 °C, and cooling water temperature 20 °C).

**Figure 15 membranes-10-00207-f015:**
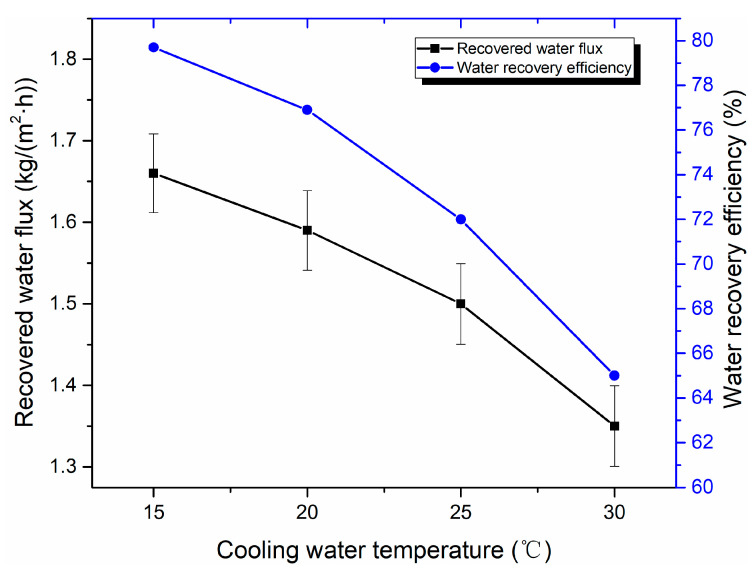
Effect of cooling water temperature on water recovery performance (Experiment conditions: flue gas flow rate of 9 L/min, flue gas temperature of 50 °C, and cooling water flow of 1 L/min).

**Table 1 membranes-10-00207-t001:** Chemical composition of ceramic raw materials.

Material	SiO_2_	Al_2_O_3_	Fe_2_O_3_	CaO	MgO	Na_2_O	K_2_O	SO_3_	Loss of Ignition
Fly ash	50.6	27.1	7.1	2.8	1.2	0.5	1.3	0.3	8.2
Talcum	58	20	/	1.8	18	/	/	/	/

**Table 2 membranes-10-00207-t002:** Raw materials of ceramic membrane supports.

Materials	Specifications	Sources
Fly ash	100 Mesh	A power plant of China Datang corporation
Talcum	3000 Mesh industrial grade	Shijiazhuang, Hebei
Carboxymethylcellulose	Industrial grade high viscosity	Cangzhou, Hebei
Dextrin	Industrial grade high viscosity	Ji’nan, Shandong
Glycerin	99% Purity	A chemical plant in Beijing

**Table 3 membranes-10-00207-t003:** Parameters of experimental apparatus.

Experimental Apparatus	Model	Parameters	Precision	Manufacturer
Gas flow controller	D07-9E	30 SLM	±2%	Beijing Sevenstar, China
Temperature and humidity transmitter	TH-21E	Temperature Range: −40 °C to 125 °C RH Range: 0–100%	≤±0.2 °C ≤±2%	Guangzhou Anymetre, China
Eight-loop digital display device	HT-MK807-01-23-KL	/	0.5% FS	Hantang Precision Instrument, Shanghai, China
Thermocouple	PT100	−50 °C to 200 °C	A Class	Hangzhou Sinomeasure, China
Miniature electric diaphragm pump	PLD-1205	Max flow rate: 3.2 L/min	/	Shijiazhaung Pulandi, China
Flow meter	LZT-M15	Range: 0.2–2.0 LPM	±4%	VAKADA, Beijing, China

**Table 4 membranes-10-00207-t004:** Comparison of different researches (Q/A: the ratio of flue gas flow rate to membrane area).

Reference	Pore Size	Component	Model	Q/A (m^3^/h/m^2^)	Water Flux kg/(m^2^·h)	Experimental Conditions
[43]	7 nm	Air/water vapor	Single tube	191	4.50	Inlet gas temperature and flow rate were 100 °C and 6.7 L/min, respectively. Cooling water flow rate was 3.3 L/h.
[20]	1 μm	Gas-fired boiler flue gas	Bundle	2286	15.77	Inlet gas temperature and flow rate were 46 °C and 1600 m^3^/h, respectively. Cooling water temperature and flow rate were 23 °C and 1150 L/h, respectively.
[47]	50 nm	Air/water vapor	Single tube	30	4.82	Inlet gas temperature and flow rate were 60 °C and 15 L/min, respectively. Cooling water temperature and flow rate were 20 °C and 1 L/min, respectively.
This paper	5–9 μm	N_2_/water vapor	Single tube	18	5.22	Inlet gas temperature and flow rate were 70 °C and 9 L/min, respectively. Cooling water temperature and flow rate were 20 °C and 1 L/min, respectively.

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
