# Peer review of "Study on the Preparation and Properties of Talcum-Fly Ash Based Ceramic Membrane Supports"

_membranes, 2020, doi:10.3390/membranes10090207_

Round 1

Reviewer 1 Report

The work is devoted to the design of inorganic porous membranes from cheap commercially available reagents. Membranes are promising for drying flue gases, since they have high chemical stability in the presence of acidic components. Thus, the work is interesting for specialists working in membrane science and technology. However, the reviewer has the following comments:

The advantages of the vapor condensation method proposed by the authors are described in the works [Journal of Membrane Science 523 (2017) 291–300; Separation and Purification Technology 171 (2016) 191–196]. Therefore, it would be good to discuss these publications in the introduction to the article.

To assess the performance of the process, the reviewer recommends using not the flux of condensed water, but the flux of dried gas (the volume of dried gas per unit of membrane area per unit of time). Gas flow is the target process variable and water flow is a secondary variable. Figures 9-12 should be discussed from this point of view.

The power needed to drive the membrane condenser process is often used to assess the efficiency of the membrane gas drying process [Separation and Purification Technology (2017), 181, 60-68.]. Since this article is devoted to applied problems, it would be good if the authors evaluate the energy efficiency of the proposed condenser.

After revision of the manuscript, the work can be published in Membranes.

Reviewer 2 Report

The paper has presented an interesting route to fabricate fly ash based membrane for application in water recovery from flue gas. Overall the paper looks fine, however, the authors should address the following comments.

Introduction: provide the spelling for the terms TFC and PSF.

Experimental: the authors mentioned that the size of fly ash particles was uniform and the particle size distribution was narrow, however, the analysis of fly ash particle size after ball milling is not provided in their results. It would also be useful to provide more information about the parameters used in the ball milling step, as well as more details about the refining and stalling steps used in the preparation of the ceramic slurry. For the sintering step, the authors indicated that “After drying, sintering was carried out according to the predetermined sintering temperature curve” but did not provide the sintering curve.

The authors wrote “The pore size of fly ash used in this experiment was measured by Laser Scattering Particle Size Distribution Analyzer (LA-960, Horiba, France).” It should be corrected to “The particle size of fly ash ….”.

For equation (1) used to measure the bending strength, the units for the parameter (strength, load, etc.) should be provided.

Section 3 should be called “Results and Discussion”

In this section, for Fig. 5 the authors stated that “A slight bulge of the diffraction peak at the small angle shows that the anorthite phase is an amorphous phase”, however, the XRD pattern for anorthite phase clearly shows several sharp peaks indicating that this phase is crystalline. Thus, the authors should revise this statement.

It would be useful to include the error bars for the data reported in Figures 8-12. I would expect that, after adding the error bars in Fig. 11, the effect of cooling water flow rate on the water recovery performance would seem very negligible or non-existent. The same remark should be considered for Fig. 12, if applicable (the scale on the y-axis might be misleading).

Reviewer 3 Report

The manuscript investigated low-cost method to prepare ceramic membrane for water recovery using fly ash from power plants. The purpose and significance of this study were well described in the introduction section. The characteristics and performance of the prepared ceramic membrane were systematically analyzed. Although the manuscript showed high performance of the investigated ceramic membrane, the manuscript can be improved considering following comments:

  1. The mechanical strength was decreased after corrosion from 25 to ~7 MPa. It is recommended to describe more the meaning the mechanical strength for the application of the prepared ceramic membrane. How much mechanical strength would be suitable for the flue gas application.

  1. Water recovery efficiency seems high considering small membrane area used in the bench scale test. It could be due to low flow rate per membrane area. However, this aspect seems described in oppositely in the manuscript. It is suggested to compare the performance of water recovery and water flux together with other research or commercial membrane to understand the performance of investigated ceramic membrane.

  1. Selectivity of the water was not conducted. Does it require additional coating for the flue gas application?

  1. The authors described that the merit of suggested preparation of ceramic membrane is low-cost. Is it possible to compare the cost advantage in number with previous methods to clarify the merit?

Round 2

Reviewer 1 Report

The authors responded to all comments. The article can be published in the Membranes.

Reviewer 3 Report

*The manuscript has been improved as addressed..